# Chemical-Assisted CO_2_ Water-Alternating-Gas Injection for Enhanced Sweep Efficiency in CO_2_-EOR

**DOI:** 10.3390/molecules29163978

**Published:** 2024-08-22

**Authors:** Pengwei Fang, Qun Zhang, Can Zhou, Zhengming Yang, Hongwei Yu, Meng Du, Xinliang Chen, Yuxuan Song, Sicai Wang, Yuan Gao, Zhuoying Dou, Meiwen Cao

**Affiliations:** 1College of Engineering Science, University of Chinese Academy of Sciences, Beijing 100049, China; fangpengwei21@mails.ucas.edu.cn (P.F.); yzhm69@petrochina.com.cn (Z.Y.); dumeng22@mails.ucas.ac.cn (M.D.); chenxinliang20@mails.ucas.edu.cn (X.C.); littlekuriboh@163.com (Y.S.); wangsicai22@ucas.ac.cn (S.W.); gaoyuan22@mails.ucas.ac.cn (Y.G.); douzhuoying22@mails.ucas.ac.cn (Z.D.); 2Institute of Porous Flow and Fluid Mechanics, University of Chinese Academy of Sciences, Langfang 065007, China; 3State Key Laboratory of Enhanced Oil and Gas Recovery, Beijing 100083, China; yhongwei@petrochina.com.cn; 4Research Institute of Petroleum Exploration & Development, PetroChina, Beijing 100083, China; 5State Key Laboratory of Heavy Oil Processing and Department of Biological and Energy Chemical Engineering, College of Chemical Engineering, China University of Petroleum (East China), Qingdao 266580, China; zhoucan1219@163.com

**Keywords:** CCUS, CO_2_-EOR, CO_2_-WAG, heterogeneous and fractured reservoirs

## Abstract

CO_2_-enhanced oil recovery (CO_2_-EOR) is a crucial method for CO_2_ utilization and sequestration, representing an important zero-carbon or even negative-carbon emission reduction technology. However, the low viscosity of CO_2_ and reservoir heterogeneity often result in early gas breakthrough, significantly reducing CO_2_ utilization and sequestration efficiency. A water-alternating-gas (WAG) injection is a technique for mitigating gas breakthrough and viscous fingering in CO_2_-EOR. However, it encounters challenges related to insufficient mobility control in highly heterogeneous and fractured reservoirs, resulting in gas channeling and low sweep efficiency. Despite the extensive application and research of a WAG injection in oil and gas reservoirs, the most recent comprehensive review dates back to 2018, which focuses on the mechanisms of EOR using conventional WAG. Herein, we give an updated and comprehensive review to incorporate the latest advancements in CO_2_-WAG flooding techniques for enhanced sweep efficiency, which includes the theory, applications, fluid displacement mechanisms, and control strategies of a CO_2_-WAG injection. It addresses common challenges, operational issues, and remedial measures in WAG projects by covering studies from experiments, simulations, and pore-scale modeling. This review aims to provide guidance and serve as a reference for the application and research advancement of CO_2_-EOR techniques in heterogeneous and fractured reservoirs.

## 1. Introduction

In recent years, the issue of global warming caused primarily by CO_2_ emissions has become increasingly severe, making CO_2_ emission reduction a globally significant concern [1]. CCUS technology is a critical breakthrough in addressing this issue. Undoubtedly, among the numerous CCUS projects, CO_2_-EOR is a crucial method for achieving both the utilization and sequestration of CO_2_. Generally, CO_2_-EOR can increase the oil recovery factor by 8–15%, and 1 m^3^ of additional oil needs 2.4–3 tons of CO_2_ injected [2,3]. Due to the low viscosity of gas and the significant density difference between the gas and crude oil reservoir, gas injection processes exhibit poor microscopic sweep efficiency, resulting in oil bypassing, fluid front instability (e.g., viscous fingering), and an early breakthrough in the swept area of a reservoir [4,5]. WAG flooding can effectively reduce the mobility of CO_2_ by decreasing its relative permeability and partially divert gas by using water to fill flow channels previously invaded by CO_2_, thereby enhancing macroscopic sweep efficiency and oil recovery [6]. Since the first WAG project was reported in the North Pembina field, Canada, in 1957, this technique has been successfully implemented in numerous fields. More than 60 field experiences have confirmed an EOR increase of 5 to 10% over water flooding by WAG processes [7]. Skauge et al. [8] reviewed 59 WAG fields and found that the average oil recovery increases by up to 10% of the original oil in place (OOIP) for all WAG cases. The WAG performance is highly affected by the injection strategies (e.g., injection well pattern, WAG ratio, number of WAG cycles, volume of each cycle, and injection rate and pressure). Different optimal strategies in terms of recovery factors and economic aspects were found in the literature. The simulation results reported in the literature indicate that multiple WAG cycles at a WAG ratio of 1:1 can effectively enhance the sweep efficiency, achieving optimal oil recovery [9]. The weak performance of WAG in some projects can be related to inappropriate WAG parameters such as the number of cycles, the volume of each cycle, and the injection rates of the gas and water phases. Hence, WAG optimization is a proper scheme to control gas mobility and increase oil recovery [10,11]. These optimal parameters can vary with the reservoir rock and fluids characteristics [12]. However, due to gravity segregation and limited control over water–gas mobility, the efficiency of the displacement sweep in reservoirs with high permeability contrasts and fractured heterogeneous formations are markedly diminished [13,14]. Researchers have introduced various modifications to the WAG process by adjusting the compositions of the liquid and gas phases and operational conditions to enhance its production characteristics. For instance, Stephenson et al. [15] conducted a pilot study in the Joffre Viking oilfield to evaluate the effects of a continuous CO_2_ injection, CO_2_-WAG injection, and simultaneous water and gas injection on the sweep efficiency and recovery rates. The results indicated that a simultaneous injection of water and CO_2_ at a 1:1 ratio significantly enhanced the sweep efficiency and recovery rates compared to conventional CO_2_-WAG and continuous CO_2_ injection methods. Furthermore, chemical-assisted WAG is regarded as an innovative technique that can expand the sweep volume and enhance oil recovery. This approach not only seals preferential flow paths such as fractures and large channels, but also enhances early CO_2_ breakthrough control and mitigates viscous fingering during WAG [16,17]. Faizal et al. [18] demonstrated that injecting gas-phase foam during WAG can adjust the mobility ratio, thereby reducing the gravity segregation and viscous fingering of water and gas. The injected foam can seal high-permeability channels, potentially increasing the recovery rate by 5% compared to conventional WAG injection methods. Moreover, adding polymers to the water slug or using gels can seal large fractures and preferential flow paths, thereby improving the mobility ratio and enhancing the sweep efficiency accordingly. For instance, Li et al. [19] conducted numerical simulations on the North Burbank Unit TR59 to study the feasibility and performance of polymer-enhanced WAG in highly heterogeneous reservoirs. They found that polymer-enhanced WAG yielded an oil recovery of 20%, surpassing conventional WAG by 12%. Moreover, in heterogeneous reservoirs with a permeability exceeding 500 mD, polymer-enhanced WAG processes achieved 7–15% higher recovery rates compared to conventional WAG processes.

To the best of our knowledge, the literature on CO_2_-WAG processes lacks a comprehensive review encompassing the effects of reservoir and fluid properties, operational conditions, process mechanisms, and advancements in WAG improvement techniques on the volumetric sweep efficiency across multiple scales, both theoretically and experimentally. This review aims to comprehensively synthesize the extensive research on the CO_2_-WAG sweep efficiency available in the literature for the first time. This study investigates the impact of key factors, including reservoir properties, fluid properties, and operational conditions, on CO_2_-WAG sweep efficiency using experimental simulations, modeling, and pore-scale analyses. Furthermore, this paper presents proposed regulatory strategies to mitigate these issues. This review includes a brief explanation of the mathematical modeling of the WAG process, followed by a systematic discussion on the significant influence of two-phase and three-phase hysteresis models on the development of WAG mathematical models. Additionally, we discuss the primary mechanisms, controlling factors, and future directions of the latest advancements in physical and chemical-assisted CO_2_-WAG techniques for enhancing the sweep efficiency. Finally, we address the technical and non-technical challenges associated with CO_2_-WAG, followed by conclusions and recommendations.

## 2. Key Factors Influencing Sweep Efficiency in WAG Flooding

The performance of flooding processes such as gas and a WAG injection is highly dependent on the macroscopic volumetric and microscopic displacement efficiencies [20]. Displacement efficiency refers to the oil produced from the pore spaces by the injecting fluid, while volumetric sweep efficiency indicates the amount of oil produced that has come into contact with the injected fluid [20]. The total oil recovery efficiency results from the combination of both displacement efficiency (*E_d_*) and volumetric efficiency *(E_v_*) [21], as shown below:(1)$$E=E_{d}\times E_{v}$$
where *E* is the total recovery efficiency (%) and *E_d_* is influenced by the mobility ratio and capillary forces. An optimal mobility ratio can enhance the effective movement of displacing and displaced fluids within the reservoir, leading to a stable and consistent displacement front. The mobility ratio is defined as follows:(2)$$M=\frac{\lambda_{1}}{\lambda_{2}}$$
where *λ*_1_ is the mobility of the displacing fluid (injected water/gas) and λ_2_ resembles the mobility of the displaced fluid (e.g., oil). The mobility ratio affects both the macroscopic volumetric and microscopic displacement efficiencies. The capillary number (*N_ca_*) is given by [22]:(3)$$N_{ca}=\frac{\nu \mu }{\sigma }$$
in which *σ* is the interfacial tension, IFT (N/m), *µ* refers to the viscosity of the displacing fluid (Pa·s), and *υ* is the Darcy velocity (m/s). One of the strategies to increase the capillary number is through a reduction of the interfacial tension, using a surfactant or thermal energy. In a miscible displacement process, the capillary number becomes infinite and the residual oil saturation in the gas-swept region may approach zero upon a favorable mobility ratio [23]. The *E_v_* of a CO_2_-WAG injection is predominantly influenced by several factors, such as the petrophysical properties of the reservoir, fluid characteristics, field-scale considerations, and economic aspects [24]. Through a comprehensive literature review, we have examined the effects of reservoir heterogeneity, relative permeability, hysteresis, and field injection-production regimes on the sweep efficiency of a CO_2_-WAG injection, particularly focusing on petrophysical properties and production strategies.

### 2.1. Mobility Ratio

Since the viscosity of CO_2_ ranges from 0.05 to 0.1 cP, which is much lower than the viscosity of formation water, an unfavorable water–gas mobility ratio during the gas injection often leads to viscous fingering, resulting in premature breakthrough and thus reducing the sweep efficiency and recovery factor [25]. A WAG injection can regulate the water–gas mobility ratio, thereby enhancing the sweep efficiency. By alternating the injection of water and gas into the rock pores, the gas-phase flow resistance is increased, forcing the gas to divert into smaller pore throats not previously swept by water, thereby suppressing gas channeling and expanding the sweep volume. Furthermore, the addition of chemical agents (nanoparticles, gels, or polymers) during the WAG injection can more effectively control the water–gas mobility ratio [16,17,18]. This approach can block preferential flow paths such as fractures and large pores, and it is also a crucial method for improving early CO_2_ breakthrough and viscous fingering during the WAG injection. As shown in Figure 1, Afzali [17] demonstrated that adding gels to the water phase of a WAG injection (to control the mobility ratio) enables CO_2_ to reach reservoir areas that traditional WAG cannot. Additionally, the increased gas-phase viscosity improves CO_2_ displacement by reducing viscous fingering and gas channeling. The numerical simulation results indicate that adding gels to the water phase during a WAG injection can increase oil recovery by 7–8% compared to a conventional WAG injection [26].

### 2.2. Reservoir Properties and Gravity Segregation

Factors affecting the vertical sweep volume in WAG flooding include reservoir properties such as the dip angle, permeability, heterogeneity, and porosity variations, as well as the gravitational segregation of the gas [27,28]. To more clearly explain how reservoir properties and gravitational segregation control the vertical sweep volume, we introduce the viscous/gravity dimensionless number (*Rv*/*g*), defined as follows [29]:(4)Rvg=vuokog∆ρLh
where *v* is the Darcy velocity, *μo* is the oil viscosity, *L* refers to the distance between the wells, *k_o_* is the effective oil permeability, *g* denotes the acceleration due to the gravity, Δ*ρ* is the density difference between the fluids, and *h* represents the height of displacement zone.

Generally, gas tends to migrate to the top of the reservoir due to gravitational segregation. As the vertical sweep height (*h*) increases, the vertical sweep efficiency improves, thereby enhancing recovery rates [30]. However, the vertical gas sweep is often hindered by reservoir properties such as the dip angle, permeability, heterogeneity, and porosity variations, preventing the gas from reaching the top of the reservoir. For example, as porosity or permeability increases, the fluid flow tends to stabilize, leading to improved displacement efficiency [31]. As illustrated in Figure 2, during a WAG injection, a gas injection can typically penetrate the upper portions of rhythmic reservoirs characterized by favorable physical properties and homogeneity, which water flooding cannot reach. This process enhances the vertical sweep efficiency and displaces the residual oil at the top of the reservoir. However, studies have indicated that during CO_2_ flooding, the production dynamics of oil wells are significantly influenced by reservoir heterogeneity [32]. In highly stratified reservoirs, the displacement front tends to move along highly permeable layers, thereby bypassing significant amounts of residual oil in less permeable layers [33]. This impact is more pronounced in reservoirs with varying vertical permeability gradients, particularly in anti-rhythmic reservoirs. Due to gravity segregation and a reduced displacement speed, the advancement of the displacement front in low-permeability layers is restricted, potentially reducing the sweep range of the gas injection and decreasing oil recovery. Generally, the greater the reservoir heterogeneity, the smaller the sweep volume in WAG flooding [34,35,36,37].

### 2.3. Hysteresis and Relative Permeability

The swept volume of a WAG injection is intricately linked to the hysteresis effects caused by the entrapment of the non-wetting phase CO_2_ in the rock’s pore spaces. Hysteresis occurs due to the trapping of the non-wetting-phase gas in various forms, such as slugs or bubbles, within the rock pores, leading to delayed responses in the relative permeability and capillary pressure [38]. During WAG injection cycles, the relative permeability of the non-wetting phase exhibits significantly more pronounced hysteresis. In the imbibition process, the relative permeability of the non-wetting phase is substantially lower compared to the drainage process, reducing gas mobility, delaying gas breakthrough, and increasing the sweep efficiency of the gas drive [39,40].

Furthermore, during the initial water injection (imbibition) process, the wetting phase water, driven by capillary forces, will first fill the smaller rock pores and displace the oil within these pores. As the frequencies of water injections and water saturation increase, the two thin water films on either side of the rock throats will gradually connect, trapping the oil between the water films in the throats (Figure 3a). With the continued high-pressure injection of water and gas, the water or gas phase increasingly tends to bypass the water-blocked regions containing residual oil, which affects the sweep efficiency of a WAG injection. Foroudi [41] suggests that as the contact angle increases (weaker water-wet conditions), water and gas bypassing become the primary mechanism for displacing oil during a WAG injection. High initial water saturation in the reservoir accelerates the formation of water-blocked regions, trapping more residual oil within the rock pores and exacerbating water and gas bypassing (Figure 3b). Karkooti et al. [42], through numerical simulations of offshore oil fields in Malaysia, also found that a high-pressure water or gas injection tends to bypass the water-blocked regions and promote the formation of three-phase-flow bypassing patterns.

### 2.4. WAG Field Injection-Production Scheme

Extensive WAG experiments and field applications, both domestically and internationally, have demonstrated that the sweep efficiency of WAG flooding is closely related to operational parameters, including the water–gas ratio, slug size, injection pressure, injection rate, and injection method [22,43]. Generally, in most oil field studies, the optimal water–gas ratio is found to be 1:1 due to its higher recovery rates, with small slugs (0.1–0.3 PV) proving the most effective, this consensus aligns with prior laboratory experiments and numerical simulations [44,45]. An injection below or above the optimal velocities is detrimental, potentially causing premature gas breakthrough and channeling [46]. Maintaining the injection pressure within an appropriate range is crucial for an effective CO_2_-WAG injection. An abnormal increase in the injection pressure or MMP of CO_2_ and crude oil can diminish the CO_2_-WAG sweep efficiency and the extent of the miscible zone. Choosing the optimal injection pattern is pivotal in WAG project design. This entails expanding the sweep volume to enhance oil displacement. Several vital factors that should be considered in designing an optimal injection pattern include reservoir heterogeneity, directional permeability, fracture directions, the physical properties of the injected fluids, well intervals, and the performance of injection and production wells [47]. A review of the WAG field cases reports that the five-spot injection pattern is the most common configuration in onshore fields [46].

Studies [47] have shown that optimizing the WAG injection process can mitigate poor volumetric sweep efficiency resulting from reservoir heterogeneity and gravity segregation effects. Surguchev [48] found experimentally that the vertical sweep efficiency in WAG processes significantly depends on reservoir anisotropy and the ratio of viscous to gravitational forces. Vertical reservoir heterogeneity and gas gravity segregation reduce the stability of lateral flow velocities and displacement fronts, thereby negatively impacting lateral sweep volumes and recovery rates. Controlled adjustments of the water–gas ratio and injection rates during WAG processes can increase gas injection velocities and displacement fronts in both low-permeability and high-permeability layers, thereby enlarging gas sweep volumes. Ding et al. [49] demonstrated through numerical reservoir simulations using three injection methods—a continuous CO_2_ injection (CGI), constant water–gas ratio WAG (CWAG), and decreasing water–gas ratio WAG (TWAG)—the TWAG injection effectively mitigates gas breakthrough and viscous fingering issues caused by gas gravity override and vertical reservoir heterogeneity. This method enhances CO_2_ utilization efficiency and petroleum recovery rates.

As shown in Table 1, considering both the ease of control and implementation sequence, reducing the water–gas mobility ratio is a straightforward and effective method to mitigate CO_2_ gas channeling and viscous fingering. This can be accomplished by introducing chemical agents to physically or chemically block fractures and high-permeability channels, thereby enhancing reservoir heterogeneity and adjusting the water–gas mobility ratio. Additionally, selecting optimal production and injection regimes can enhance gas lag effects and mitigate the adverse impacts of gas bypassing and gravity segregation, which reduce the gas sweep efficiency. This approach enhances the WAG flood sweep efficiency and boosts oil recovery rates.

## 3. Establishment of WAG Flooding Mathematical Models and Hysteresis Models

The governing equations for an immiscible three-phase flow in porous media (with an application to the WAG process) use the classical formulation of Muskat’s extension [50] for Darcy’s equation. The continuity and auxiliary equations are given in this section. The continuity equation for phase α is as follows [33]:(5)$$\frac{\partial \left(\phi \rho_{i}S_{i}\right)}{\partial t}+\nabla \cdot \left(\rho_{i}u_{i}\right)=q_{i}\;;\;i=w,o,g$$
where *t* stands for time; *ϕ* resembles the medium porosity; *ρ_i_* is the density of phase *i* (which can be water, oil, or gas); *S*_i_ is the saturation of phase *i*; *u_i_* is the velocity of phase *i*; and *q_i_* denotes the sink or source term. The velocity of phase *i* can be obtained from the extended Darcy’s equation for the multiphase flow systems as given below [51]:(6)$${\;u}_{i}=\frac{Kk_{ri}}{\mu_{i}}\left(\nabla p_{i}-\rho_{i}g\right)\;;\;i=w,o,g$$
in which *K* is the intrinsic permeability of the porous medium; *k_ri_* represents the relative permeability of phase *i*; *p_i_* is the pressure of phase *i*; *μ_i_* is the viscosity of phase *i*; and g denotes the acceleration vector due to gravity.

The auxiliary flow correlations are shown below:(7)$$k_{r\alpha }=k_{r\alpha }(S_{w},S_{o},S_{g})$$
(8)$$p_{cow}(S_{w},S_{o},S_{g})=p_{o}-p_{w}$$
(9)$$p_{cgo}(S_{w},S_{o},S_{g})=p_{g}-p_{o}$$
(10)$$p_{cwg}(S_{w},S_{o},S_{g})=p_{g}-p_{w}$$

The summation of saturations of different phases are constrained:(11)$$S_{w}+S_{o}+S_{g}=1$$

Only two of the three saturation terms are independent. The three-phase relative permeability and capillary pressure models depend on phase saturations. Due to the cyclic nature of WAG processes, researchers [52] recommend using three-phase relative permeability and capillary pressure models that incorporate hysteresis effects. Moreover, studies [41,53] have demonstrated that neglecting hysteresis effects in WAG process simulations can result in significantly lower predicted recovery rates compared to models that consider hysteresis. Therefore, investigating hysteresis effects is crucial for developing accurate mathematical models for WAG flooding. We investigated how hysteresis affects WAG production and common models of relative permeability hysteresis.

The phenomenon of hysteresis in petroleum engineering has long been recognized. Hysteresis refers to irreversibility or path dependency, and for flow in porous media, it manifests through the dependence of relative permeability and the capillary pressure on the saturation path [38]. From the perspective of pore-scale processes, there are at least two sources of hysteresis [54]. The first source is contact angle hysteresis. Due to surface roughness or chemical heterogeneity, the advancing contact angle (when the wetting phase displaces the non-wetting phase) is greater than the receding contact angle (when the wetting phase is displaced by the non-wetting phase). The second source is the trapping of the non-wetting phase. For reservoir rocks with a strong preference for wetting, when the saturation changes from drainage (decreasing wetting phase saturation) to imbibition (increasing wetting phase saturation), the non-wetting phase can be trapped by the wetting phase. Specifically, during a cyclic water and gas injection in WAG processes in porous media, if the saturation of the wetting phase (water) rapidly increases during the imbibition process in hydrophilic rocks, the non-wetting phase (gas) can be dispersed by the wetting phase, becoming discontinuous and trapped in various forms such as slugs and bubbles within the rock pores. This trapping (a fixed amount of the non-wetting phase) leads to hysteresis in relative permeability [39].

It is well known that relative permeability is not only a function of saturation, but also a function of the saturation history/path [55]. Relative permeability can be regarded as an overall parameter influenced by various factors during the water-alternating-gas (WAG) process, including rock wettability, chemical influences (such as salinity and pH), saturation, fluid viscosity, flow rate, pore size distribution, and IFT [56]. In fact, relative permeability represents the petrophysical relationship most relevant to the WAG process and is also a fundamental physical quantity in two-phase or three-phase hysteresis models. Extensive research has been conducted by scholars both domestically and internationally on the phenomenon of relative permeability hysteresis. Geffen et al. [57] studied the effect of the saturation history on relative permeability boundary curves in gas–water systems. Through experiments, they concluded that the relative permeability of the two phases is not a single-valued function of saturation, and for the water–gas (air) system, the hysteresis effect is more pronounced for the gas phase compared to the water. Batycky [58] discovered through experiments that relative permeability hysteresis is related to saturation and interfacial tension. Subsequently, Jr. Ertekin et al. [59,60] experimentally confirmed that relative permeability hysteresis is also related to the capillary number and pore structure. Morrow et al. [61,62] found that reservoir rock wettability plays an important role in the relative permeability and its hysteresis behavior. Based on the assumption of strongly water-wet porous media, Land [63] was the first to define “flow saturation” and established a basic trapping model estimating the imbibition relative permeability at a given actual saturation using the drainage relative permeability at flow saturation. Building on Land’s relative permeability hysteresis model, Killough [64] suggested that each hysteresis curve can be determined by the points of the saturation direction reversal on the imbibition and drainage boundary curves. Carlson [65] proposed a hysteresis model for the relative permeability of the non-wetting phase, assuming no hysteresis effect for the wetting phase. Theoretically, the imbibition curves should be parallel to each other. However, experiments by researchers such as Delshad et al. [66,67] have shown that many reservoirs are mixed-wet. Kjosavik [68] proposed a hysteresis model for mixed-wet rocks. They extended the relative permeability relationships used for water-wet and oil-wet systems to mixed-wet rocks by using a weighted method with capillary pressure data. Subsequently, Spiteri et al. [69,70,71] demonstrated through experiments and the introduction of hysteresis models that relative permeability hysteresis can significantly impact recovery rates during the WAG process. To this end, we have reviewed the existing literature and investigated common two-phase and three-phase hysteresis models, discussing their respective advantages, disadvantages, and applicable scenarios.

### 3.1. Two-Phase Relative Permeability Hysteresis Models

The initial step in characterizing relative permeability hysteresis in two-phase systems accurately calculates the trapped saturation of the non-wetting phase for any displacement path. Calculating the trapped saturation of the non-wetting phase is essential for any hysteresis model, as it determines the endpoint saturation of the oil and gas relative permeability curves [55,64]. Extensive experimental and theoretical studies have focused on elucidating the trapping mechanisms that control multiphase flow in porous media. Land’s model is the most widely used hysteresis model. This single-parameter model forms the foundation of numerous relative permeability hysteresis models. Land’s model [63] assumes that the wetting-phase relative permeability scanning curves are reversible, the trapped saturation of the non-wetting phase (gas in his model) has a monotonic relationship with the initial gas saturation, and the Land trapping factor (*C*) changes between hysteresis cycles as illustrated below.
(12)$$C=\frac{1}{S_{gt}}-\frac{1}{S_{gi}}$$
where *C* represents the Land trapping factor, *S_gt_* denotes the trapped gas saturation, and *S_gi_* stands for the initial gas saturation [51,72]. In addition to the Land trapping model, other two-phase hysteresis models exist, most of which are improvements and optimizations of the Land trapping model. Table 2 enumerates the common two-phase hysteresis models and their main equations. Each of these models is suited to specific systems and their corresponding rock types. However, their applicability to intermediate-wet and mixed-wet systems remains limited.

### 3.2. Three-Phase Relative Permeability Hysteresis Models

Unlike two-phase systems, a three-phase flow involves at least two independent saturations, resulting in infinitely many possible flow paths for saturation distribution. Consequently, measuring the relative permeabilities of all possible three-phase flows at every production stage in a reservoir is impractical and expensive [74]. To date, interpolating three-phase relative permeabilities from two-phase experimental data (oil–water, oil–gas, and gas–water) has emerged as a practical solution to this problem. Specifically, three-phase relative permeability models utilize two-phase relative permeability measurements to estimate the relative permeability of the oil phase in a three-phase system. These models assume that the relative permeabilities of the wetting phase (water) and the non-wetting phase (gas) depend solely on their respective saturations [55]. Therefore, their relative permeabilities are functions of their own saturations [75], expressed as *k_rw_* = *k_rw_*(*S_w_*) and *k_rg_ = k_rg_*(*S_g_*). The relative permeability of the oil phase is a function of both water and gas saturations, expressed as *k_ro_ = k_ro_*(*S_w_, S_g_*).

Table 3 summarizes common three-phase hysteresis models, which derive three-phase permeabilities by interpolating two-phase permeability data and integrating them into their respective models. A common feature of Carlson and Killough’s hysteresis models, as well as other two-phase hysteresis models, is the assumption of the reversibility of relative permeability curves at the boundary between drainage and imbibition. This assumption contradicts the observed irreversibility of hysteresis scanning curves from experiments [64,65]. Currently, two-phase relative permeability hysteresis models do not adequately and accurately describe hysteresis in WAG cycles, which is because most models predict imbibition hysteresis curves under different initial conditions rather than hysteresis curves for drainage and imbibition within cyclic periods. Thus, three-phase hysteresis models derived by coupling two-phase hysteresis models, such as the Stone 1 and Stone 2 models, also fail to accurately predict three-phase hysteresis permeabilities in the WAG process [76].

## 4. Techniques to Expand CO_2_-WAG Sweep Volume

At present, a traditional CO_2_-WAG injection can increase the sweep efficiency and enhance oil recovery by controlling the mobility ratio, but it still faces issues of gas channeling and low gas sweep efficiency in highly heterogeneous reservoirs [81,82]. Improved water-alternating-gas techniques can effectively overcome the weak mobility control of a traditional WAG injection, enhancing the sweep volume of CO_2_. These methods for controlling CO_2_ gas channeling have been proven effective in both field applications and laboratory simulations. The main objective of this section is to elucidate the sealing mechanisms by different improved CO_2_-WAG techniques and the parameters affecting their effectiveness.

### 4.1. Simultaneous Water and Gas Injection (SWAG) for Enhanced Sweep Volume

As illustrated in Figure 4, the SWAG injection technique leverages the gravity differences between water and gas during a separate advancement, achieving a vertical sweep efficiency unattainable by a single displacing phase. The presence of the water phase accelerates gas trapping and retention in large channels, forming micro-scale blockages in the pores, thus increasing the flow resistance in these large channels. Consequently, this forces the gas to be redirected into smaller pore throats unswept by the water drive, effectively preventing gas breakthrough and increasing the sweep volume [83,84].

Compared to a WAG injection, SWAG injection technology provides enhanced flow control by stabilizing the gas displacement front, thereby improving the sweep efficiency and oil recovery rates [85,86]. Table 4 summarizes recent research progress on SWAG technology for expanding the sweep volume. Jamshidnezhad et al. [82] conducted oil displacement experiments using a five-spot well pattern on-site, revealing a sweep efficiency of 90% with a SWAG injection, significantly higher than the 60% efficiency achieved with a gas injection alone.

The research indicates [86,87] that the capability of a SWAG injection to expand the sweep volume is influenced by reservoir heterogeneity and on-site injection-production system processes. M. Jamshidnezhad et al. [82] conducted sensitivity analyses on injection-production design parameters for SWAG performance using numerical simulations on homogeneous and heterogeneous reservoirs. They found that factors such as the injection rates of water and gas, injection well placement, and horizontal and vertical reservoir heterogeneity significantly affect the gas mobility and sweep range, with a minimal impact on the recovery rates. In evaluating the impact of on-site injection-production processes on CO_2_-WAG, continuous CO_2_, and SWAG oil displacement performances, Shrinidhi Shetty et al. [89] conducted core displacement experiments with four different gas flow rates (*fg* = 0.2, *fg* = 0.4, *fg* = 0.6, and *fg* = 0.8). They observed that, except for *fg* = 0.8, all SWAG drives exhibited superior gas utilization efficiency compared to gas drive and WAG, highlighting the ability of SWAG to control gas mobility effectively and enhance sweep efficiency for contacting residual oil in smaller pores.

In summary, while SWAG technology represents a viable method for expanding the gas sweep volume, it has drawbacks, such as high equipment and operational costs, and complex well-completion designs. Moreover, instability at the gas front, gravitational segregation of water and gas, and a CO_2_ injection can increase the corrosion of the casing and integrity issues in SWAG operations. Therefore, preventing gas breakthrough is crucial [12,90]. However, advancements in stabilizing the gas front, preventing gas breakthroughs, mitigating hydrate-induced blockages, and reducing casing corrosion could significantly enhance the applicability of SWAG technology.

### 4.2. Foam-Assisted WAG (FAWAG) for Enhanced Sweep Efficiency

Foam is a gas dispersion system surrounded by a liquid film, prepared and stabilized by a surfactant [91,92]. The primary principle of FWAG technology involves alternately or simultaneously injecting gas and a foam solution containing surfactants into the formation to generate in situ foam (Figure 5). The Jamin effect gradually increases flow resistance, reduces gas phase mobility, and redirects gas into smaller pores, thereby delaying gas breakthrough and improving sweep efficiency [93,94,95]. Additionally, a foam injection can supplement reservoir energy and reduce the gas–oil ratio, significantly enhancing oil recovery [96]. The foam injection methods in the FAWAG process include a pre-foamed injection, foam co-injection, and surfactant-alternating-gas (SAG) foam. Injected foams can be categorized by function into three types: mobility control foam (MCF), blocking/diverting foam (BDF), and gas–oil ratio control foam (GOR) [97].

Compared to a conventional WAG injection, FAWAG technology can effectively address the gravity override caused by the density differences between water and gas and formation leakage. By creating foam-blocking barriers to prevent upward gas migration, it forces the gas to redirect into smaller pores and laterally disperses, thereby displacing previously unswept residual oil [98]. R. Kharrat [99] conducted an analysis to study the impact of different injection and production modes on recovery rates in a highly fractured carbonate oilfield in the Middle East. Displacement experiments and numerical simulations revealed that FAWAG technology achieved higher recovery rates compared to GI, WAG, SWAG, and Gas-Assisted Gravity Drainage (GAGD) techniques. The primary reasons were that foam reduces interfacial tension (IFT) and capillary forces between reservoir fluids, increases the sweep volume, and decreases residual oil saturation. Similarly, Tunio S.Q. [98] conducted a comparative study of FAWAG and SWAG oil recovery performances in a Malaysian oilfield. The study found that both SWAG and FAWAG effectively control gas mobility, gravity segregation, gas channeling, and viscous fingering, thereby improving the sweep efficiency. However, in terms of the final recovery rates, SWAG increased recovery to 88%, while FAWAG significantly boosted it to 92%. C. Zhang et al. [100] addressed the common issues of early gas breakthrough and channeling during CO_2_ flooding in heterogeneous reservoirs, where injected CO_2_ forms flow channels along high-permeability zones, leaving low-permeability zones untapped. Through optimization experiments on different CO_2_ injection methods and design parameters (such as the gas–water ratio and slug size), they found that compared to continuous CO_2_ injections, CO_2_-WAG, and CO_2_ foam flooding, FAWAG not only provides the blocking and profile modification properties of foam slugs, but also maintains foam stability, making it the most effective injection method among the four technologies.

FWAG technology is an important method for addressing gas channeling and viscous fingering issues in heterogeneous reservoirs. However, foam, as a thermodynamically unstable system, interacts with the porous media of rocks through friction and physicochemical processes. Additionally, foam decays and collapses when encountering high-salinity and high-temperature formation water or crude oil, making it prone to breakdown. Therefore, the stability of the foam is a key factor affecting the efficacy of the FAWAG process [101,102]. Studies have shown [103,104] that injecting low-salinity water (i.e., brine with a modified ionic composition) helps form more stable foam films. To enhance foam stability, Anas. M. Hassan et al. [105] proposed a novel enhanced oil recovery technique. The basic principle is to combine smart water (a modified injection of water with an ionic composition and concentration to achieve optimal wettability and foam stability) with FAWAG to improve foam stability and expand the gas sweep volume. Hassan A. [106] utilized the numerical simulation software CMG (STARS module)and conducted displacement experiments on carbonate cores. He found that combining smart water with a foam system effectively controlled gas channeling and viscous fingering, thereby improving the sweep efficiency. This led to an increase in the original oil in place by 42% and a final cumulative oil production increase of 92%.

In addition to the stabilizing effects of smart water on foam, research has found [107,108,109] that nanoparticles can synergistically stabilize foams with surfactants, enhancing the liquid film strength and reducing internal foam rupture rates. This enhancement addresses the poor stability issues of CO_2_ foam systems in oil recovery. Furthermore, gels also act as foam stabilizers; foams coated with gels can improve the mechanical strength of foam interfaces and enhance foam stability. Wang et al. [110], through experiments employing a two-dimensional packed sand tube model, investigated the effect of adding nanoparticles on the thermal and oil resistance properties of CO_2_ foam. They found that silica nanoparticles adsorbed at the gas–liquid interface enhance the mechanical strength of CO_2_ foam, thereby increasing its stability compared to conventional CO_2_ foam. Yang et al. [111] conducted screening experiments on gel foam systems to address severe gas channeling in CO_2_-enhanced oil recovery. The experimental results demonstrate that foam gel systems can effectively plug low-permeability fractured cores, enhancing the foam performance, increasing the foam liquid film thickness, and effectively blocking CO_2_ gas channeling pathways.

Moreover, some CO_2_-sensitive foaming agents exhibit superior foam stability compared to conventional foaming agents. Li et al. [112] found that long-chain polyamine compounds (RDPTA) are effective CO_2_-sensitive chemicals capable of generating in situ foams triggered by CO_2_. Due to a tighter surfactant arrangement in the liquid film and inherent self-enhancement, higher CO_2_ concentrations enhance the foaming ability, resulting in more significant plugging and flow control capabilities. Therefore, leveraging these properties in experimentally simulated heterogeneous reservoirs, sequential water flooding, CO_2_ flooding, WAG, and FAWAG flooding were conducted, revealing that FAWAG flooding effectively enhances the sweep efficiency and oil recovery in low-permeability layers. Compared to conventional foaming agents, its foam system can still exert flow control even at lower CO_2_ concentrations. Thus, foams produced by the synergistic action of surfactant additives exhibit better profile control effects than single surfactant systems, with these effects closely related to the proportions and formulations of different surfactant types. This will also be a crucial direction for future FAWAG development.

### 4.3. Polymer and Gel-Assisted WAG for Enhanced Sweep Efficiency

#### 4.3.1. Polymer-Assisted WAG (PWAG) for Enhanced Sweep Efficiency

For strong heterogeneous reservoirs, large fractures and channels serve as significant pathways for gas channeling, especially those with apertures exceeding 0.5 mm. FWAG technology frequently encounters challenges in these contexts [113,114]. After a CO_2_ injection into fractured reservoirs, gas breakthrough occurs rapidly, where CO_2_ exhibits limited displacement effectiveness, resulting in a minimal sweep volume, poor oil recovery, and low extraction efficiency. Therefore, incorporating the polymer-assisted expansion of the sweep volume in WAG is essential for mitigating the formation of high-permeability channels within large fractures and channels, thereby preventing CO_2_ channeling [115].

Polymer-assisted water-alternating-gas with sweep volume expansion technology can be categorized into two main types. The first approach involves directly thickening the gas viscosity of CO_2_. Polymers, when dissolved in CO_2_, enhance its density and viscosity to varying degrees, thus mitigating issues of gas channeling caused by gravity segregation and viscous fingering [116,117]. Generally, polymers with higher molecular weights result in a greater viscosity and more effective thickening of CO_2_. However, as the molecular weight of the polymer increases, its solubility in CO_2_ decreases, which can be detrimental to CO_2_ thickening [118]. The most commonly used supercritical CO_2_ thickening agents are primarily siloxane polymers and fluorinated polymers [119,120]. Bac tested the thickening ability of polydimethylsiloxane (PDMS) on CO_2_ at 2500 psi and 54.5 °C. The viscosity of CO_2_ increased from 0.04 cp to 1.2 cp following thickening. Additionally, the use of toluene enhanced the solubility of polymers under identical pressure conditions. Bac conducted CO_2_ core displacement experiments and observed that the addition of polymers led to an increase in the oil recovery rate, delayed gas breakthrough, and an increase in oil recovery by 3.4–9% compared to its original value [121]. Additionally, Zhou et al. [122] synthesized fluorinated oligomer CO_2_ thickening agents (HMDI) and evaluated their solubility and thickening performance in CO_2_. The solubility of these agents in CO_2_ showed a good correlation with pressure. The viscosity of the system increased with the concentration of the thickening agent. When 2.0 wt% thickener was added, the system viscosity reached 19.2 mPa·s, which is 480 times higher than the initial viscosity. The second method involves introducing polymers into water to physically block gas migration pathways and regulate gas mobility. Selected polymer gels with low viscosity facilitate an effective injection, enabling the selective treatment of preferential pathways such as large conduits and fractures, thereby minimizing damage to low-permeability layers [123]. PWAG technology integrates the gas injection into small pores and polymer flooding to control fluid mobility, thereby mitigating early gas breakthrough and gravity segregation in highly heterogeneous reservoirs [124]. PWAG improves reservoir heterogeneity, minimizes gas channeling in fractures or high-permeability zones during subsequent gas displacement processes, and redirects the gas flow to displace residual oil in low-permeability and tiny pores, thereby enhancing the gas sweep efficiency and crude oil recovery. Jeong et al. [32] evaluated the oil recovery performance of PWAG processes in heterogeneous heavy oil reservoirs. Numerical simulations comparing four injection techniques (water flooding, CGI, the WAG process, and PWAG process) showed that PWAG in-creased the recovery rates by 89% compared to water flooding or continuous CO_2_ injections, and by 45% compared to the WAG process. Optimal recovery rates were achieved with a PAG-to-water–gas ratio of 2:1.

#### 4.3.2. Gel-Assisted WAG (GWAG) for Enhanced Sweep Efficiency

A CO_2_-responsive smart gel is a type of gel whose structure and properties undergo specific changes when in contact with CO_2_. By leveraging these CO_2_-responsive characteristics, the gel can be used for gas channeling control during CO_2_ flooding and for chemical sequestration during geological periods. The synthesis process of CO_2_-responsive gels primarily involves incorporating CO_2_-responsive functional groups into the molecular chains of conventional gels. As shown in Figure 6a, known CO_2_-responsive functional groups include primary amines, amidine groups, guanidine groups, tertiary amines, imidazole-containing groups, and carboxylic acids [125]. As illustrated in Figure 6b, CO_2_-responsive gels are mainly classified into pre-crosslinked and traditional CO_2_-responsive gels. In addition to the inherent properties of the gels, such as the rheology, swelling capacity, and acid and salt resistance, environmental conditions (pressure, temperature, and salinity) also significantly affect the gel performance.

Unlike PWAG technology, the GWAG technique primarily operates by incorporating CO_2_-responsive chemicals into the water to create in situ cross-linked gels. These gels chemically seal gas channels and regulate gas mobility. In the GWAG process, CO_2_ serves dual roles: as a displacing agent in enhanced oil recovery (EOR) and as a catalyst for chemical agent gelation. In practical applications, CO_2_-responsive gel properties are assessed by the CO_2_ breakthrough pressure following a gel injection to gauge sealing effectiveness, or by resistance and residual resistance coefficients to evaluate its capability in CO_2_ flow control. This paper reviews the recent research on GWAG technology, focusing on its effectiveness in expanding the CO_2_ sweep volume and mitigating gas channeling (as detailed in Table 5).

Unlike the GWAG technique, which mainly targets plugging high-permeability zones, gel multi-stage auxiliary WAG technology can seal both fractures and high-permeability layers. Zhao et al. [127] proposed this novel WAG gel-assisted technology for enhancing the sweep efficiency. It effectively controls CO_2_ gas channeling in low-permeability fractured reservoirs, thereby increasing the sweep volume and enhancing oil recovery. This system comprises two main components (as shown in Figure 7): Firstly, the injection of a high-strength starch gel solution is used to strongly seal large fractures in heterogeneous reservoirs. The high-viscosity organic amine salts produced by CO_2_ efficiently block high-permeability channels, thereby improving the heterogeneity of the reservoir’s microscopic matrix. Secondly, based on the sealing of large fractures and high-permeability channels, the subsequent gas drive sweep volume is expanded through a CO_2_-WAG injection. Building on this, Hao H et al. [128] designed a three-dimensional radial flow model experiment with complex fractures to simulate the actual oil displacement process in heterogeneous reservoirs. The experiments demonstrated that the injection of high-strength gel and ethylenediamine in the multi-stage WAG polymer auxiliary system effectively controls gas channeling along the main and micro-fractures of the three-dimensional radial model, which allows CO_2_ to penetrate the micro-pores of the rock, improving the model’s uniformity. Finally, the flow control capability of the CO_2_-WAG drive further mitigates gas channeling, forming a graded sealing system and enhancing the flow control from high- to low-permeability zones.

To evaluate the sealing effectiveness and applicability range of the combination of strong gel and small molecule amines, Xu et al. [132] conducted laboratory experiments to determine the suitable fracture scale for modified starch gel and ethylenediamine in the multi-stage WAG polymer auxiliary system. Numerical simulation experiments were also performed to simulate the fracture sealing effects. The experimental results indicated that the modified starch gel is effective for sealing fractures with an opening width of 0.42–0.65 mm, while outside this range, the sealing adaptability decreases. For medium and small fractures, ethylenediamine is suitable for sealing fractures with an opening width of less than 0.24 mm, especially those with a width of 0.08 mm. Additionally, Zhao et al. [133] conducted CO_2_ displacement improvement experiments using natural homogeneous cores and artificially made homogeneous or heterogeneous cores. These experiments aimed to investigate the applicability of different gas channeling control systems based on reservoir permeability and the fracture scale. The results demonstrated that CO_2_ displacement is suitable for heterogeneous reservoirs with a permeability ratio of 0–5; WAG displacement is appropriate for reservoirs with a permeability ratio greater than 5 and less than or equal to 30. Secondary auxiliary gas channeling control (an ethylenediamine injection) is suitable for reservoirs with a permeability ratio higher than 30 or fracture scales smaller than 0.08 mm. The tertiary auxiliary gas channeling control (an injection of modified starch gel followed by an ethylenediamine injection) exhibited excellent applicability for fracture sizes of 0.24 mm and 0.40 mm, effectively sealing macroscopic fractures and microscopic high-permeability channels, significantly increasing the sweep volume, with recovery rates exceeding 34%. However, from the current experimental results, it is found that the formulation selection indicates that modified starch gel and ethylenediamine are optimal. This system demonstrates strong selectivity towards the permeability and size of fractures, particularly suitable for fractured heterogeneous or ultra-low permeability reservoirs.

### 4.4. Nanoparticle-Assisted WAG (NWAG) for Enhanced Sweep Efficiency

Nanoparticles (NPs) are defined as materials composed of particles ranging in size from 1 nm to 100 nm [134,135]. Major NPs include Al_2_O_3_, SiO_2_, MgO, and Fe_2_O_3_, among which SiO_2_ is the most widely applied NP surfactant due to its similarity to reservoir rock components and its low cost [136].

The technology of alternating water and gas-assisted nanoparticle injections for enlarging the reservoir volume is categorized into two primary methods. The first method, NWAG, entails injecting a nanoparticle-containing aqueous solution to enhance the viscosity of the injected fluid. This process aims to improve the water–gas mobility ratio, mitigate viscous fingering, and reduce gas channeling. NPs injected into water penetrate the rock pore throats, effectively sealing high-permeability channels and modifying the wettability of reservoir rocks and the contact angle of oil–water interfaces, thereby lowering interfacial tension. Furthermore, as illustrated in Figure 8, NPs in water self-assemble into wedge structures at the three-phase interface of immiscible fluids under the combined influence of Brownian motion and electrostatic forces. Under injection pressure, these wedge films migrate through the pore spaces of porous rock, generating a separating force that dislodges oil films or droplets from the rock surface, thereby enhancing oil recovery [137]. The second method involves directly injecting gas-soluble NPs with CO_2_ (NAG) to increase the viscosity of CO_2_ and reduce asphaltene precipitation during CO_2_ flooding. This approach aims to improve the oil–gas mobility ratio and control gas channeling [138,139].

To determine the oil displacement performance and underlying mechanisms of water-based NPs, Al Matroush et al. [140] investigated the effects of water-based nanoparticles on wettability and relative permeability changes during the WAG process. The experimental results indicated that replacing water with nanofluids during the WAG process shifted the system’s wettability towards water-wet, thereby affecting the relative permeability curve. Furthermore, the reservoir simulation revealed that using a water–gas ratio of 2:1 and a five-spot well pattern increased recovery by 13% compared to traditional WAG. B. Moradi [141] introduced SiO_2_ particles into the water phase of WAG and found that SiO_2_ nanoparticles adsorbed onto the rock surface could change oil-wet rocks to strongly water-wet, reducing the oil–water interfacial tension and enhancing recovery by more than 15% compared to conventional WAG. Additionally, gas-based nanoparticles have shown promising results in controlling gas mobility and enhancing oil displacement in heterogeneous reservoirs. Zhang, K et al. [142] used CMG software 2018 to compare CO_2_ flooding, WAG, and NAG for a light oil reservoir in the Neuquen Basin. The study found that gas-based nanofluids increased CO_2_ viscosity, thus enhancing the sweep efficiency, even with modest viscosity increments, improving overall gas utilization. Moreover, the injected gas-soluble nanoparticles improved asphaltene deposition and rock wettability, leading to increased oil recovery. While NWAG techniques for enhancing sweep efficiency are characterized by a low cost and a long action distance and duration, they effectively mitigate gas channeling and viscous fingering caused by insufficient mobility control in heterogeneous reservoirs. However, NPs tend to agglomerate and generally exhibit limited adaptability to high-salinity formation water, potentially resulting in pore blockage and failure. Future research should prioritize the development of modified nanoparticles or composite systems that are compatible with high-salinity water.

## 5. Current Problems and Challenges

During WAG injections in a reservoir, the two processes of imbibition and drainage take place sequentially, resulting in complex saturation patterns as both gas and water saturations increase and decrease alternately. Hence, reliable modeling of the WAG process requires comprehensive knowledge of three-phase relative permeability and capillary pressure, including saturation directions and cyclic hysteresis effects [54]. Previous studies have shown that applying inappropriate three-phase model parameters in simulations can result in significant errors in the WAG performance [69,70,71]. Accurately predicting three-phase relative permeability data remains a challenging issue, requiring robust and reliable models or experimental outputs. WAG simulations in fractured carbonate reservoirs become computationally expensive due to multiscale heterogeneities and matrix-fracture transfer mechanisms that must be involved through detailed models with a large number of grid cells [143]. The connectivity of the fracture network and variability in the matrix structure are two sources of uncertainties in fractured carbonate flow behavior during a WAG injection [144,145]. To capture the main mechanisms involved in WAG processes, further experimental (macro and micro scales) investigations, pilot-scale tests, and pore-network modeling studies are needed to systematically conduct parametric sensitivity analyses and identify all contributing parameters with their relative importance [146]. Additionally, optimizing WAG operations in terms of process designs, operating conditions, and injected fluid properties is a challenging task requiring extensive research and engineering experience.

Additionally, in highly heterogeneous and fractured reservoirs, a conventional WAG injection struggles to meet the field’s demands for controlling CO_2_ gas channeling and enhancing reservoir heterogeneity [81,82]. Researchers have introduced various WAG techniques by modifying the liquid and gas phases, as well as operational conditions, to optimize production characteristics. Table 6 summarizes the pros, cons, and influencing factors of current chemical-assist WAG enhancement techniques. To better address gas channeling issues during CO_2_ injections, efforts should focus on not only mitigating the limitations of current technologies, but also on developing advanced CO_2_-WAG recovery techniques.

## 6. Conclusions

A CO_2_-WAG injection is a well-established CO_2_-EOR technique with a proven track record of success across numerous projects, from pore- to field-scale. Beyond its demonstrated performance, researchers have introduced various modifications to the WAG process by altering liquid- and gas-phase compositions and operational conditions to enhance its production characteristics. These modifications include physical and chemical enhancements to expand the sweep efficiency of the WAG injection, such as FAWAG, SWAG, NWAG, PWAG, and GWAG. This paper conducts an extensive study and summary of the effects of key variables (fluid properties, reservoir characteristics, and operational conditions) on the sweep efficiency of the WAG injection. We analyze existing mathematical models of the WAG injection, systematically review and summarize the impact of two-phase and three-phase hysteresis models on the development and field simulation of WAG injection models, and discuss the advantages, disadvantages, and influencing factors of different physical and chemical enhancements to WAG injections. Additionally, it identifies critical technical challenges in the field.


WAG injection technology is a crucial method for addressing significant gas channeling and viscous fingering issues. However, in the context of highly heterogeneous domestic reservoirs, it continues to face challenges, including insufficient control over gas channeling and a low sweep efficiency of CO_2_ flooding. Therefore, new and improved techniques are required to enhance the sweep volume and utilization rate of CO_2_ flooding, thereby maximizing both oil recovery and CO_2_ storage.The swept volume of CO_2_-WAG is influenced not only by its viscosity and the heterogeneity of the reservoir, but also by factors such as the mobility ratio, reservoir properties, gravity segregation, production and injection techniques, and the hysteresis effect. From the perspectives of regulatory complexity and economic viability, reducing the water–gas mobility ratio is the primary method to address CO_2_ gas channeling and viscous fingering. This can be achieved by introducing chemicals to physically or chemically block fractures and high-permeability channels, thereby improving reservoir heterogeneity and the water–gas mobility ratio. Additionally, optimizing appropriate production and injection regimes (e.g., the water–gas slug ratio, injection pressure, injection rate, and injection methods) and well pattern designs (e.g., horizontal wells and well spacing) can enhance gas hysteresis effects and mitigate the adverse impacts of gas bypassing and gravity segregation on the sweep efficiency of CO_2_ flooding. These measures collectively improve the overall utilization and storage efficiency of CO_2_ flooding.The saturation history during individual drainage and imbibition processes, along with the chronological cycle of water and gas injections in WAG, significantly influence fluid distribution in three-phase flow. Most phase permeability models for WAG flooding neglect three-phase hysteresis effects or are influenced by two-phase hysteresis models, leading to a substantial overestimation of gas phase permeability. This overestimation reduces the accuracy of realistic predictions in WAG flooding models.Chemical-assisted CO_2_-WAG for Enhanced Sweep Efficiency techniques demonstrate significant advantages over conventional CO_2_-WAG methods. They possess advantages such as enhancing the mobility ratio to mitigate gas channeling, sealing fractures, and high-permeability channels to improve reservoir heterogeneity, lowering the oil–water interfacial tension and capillary forces, and enhancing reservoir wettability. These technologies have been validated through laboratory experiments and field applications, achieving recovery rates which are more than 5% higher than those achieved with traditional CO_2_-WAG methods. With ongoing advancements in CO_2_-WAG flooding and volumetric expansion techniques, these methods are crucial for economically developing heterogeneous reservoirs and play a pivotal role in geological storage.


## Figures and Tables

**Figure 1 molecules-29-03978-f001:**
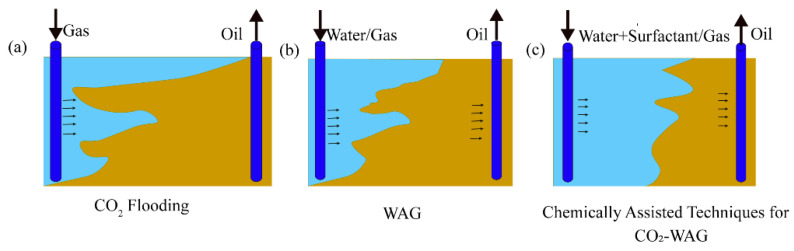
Schematic of gas channeling due to differences in mobility ratio under various injection methods, (**a**) continuous gas injection, (**b**) conventional WAG injection, and (**c**) chemical-assisted WAG. Modified after Afzali S [17], Copyright 2018, with permission from Elsevier.

**Figure 2 molecules-29-03978-f002:**
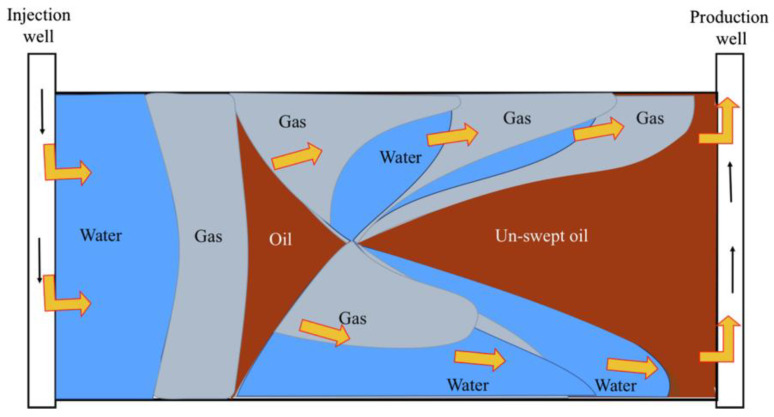
Schematic of WAG sweep efficiency influenced by reservoir properties and gravity segregation.

**Figure 3 molecules-29-03978-f003:**
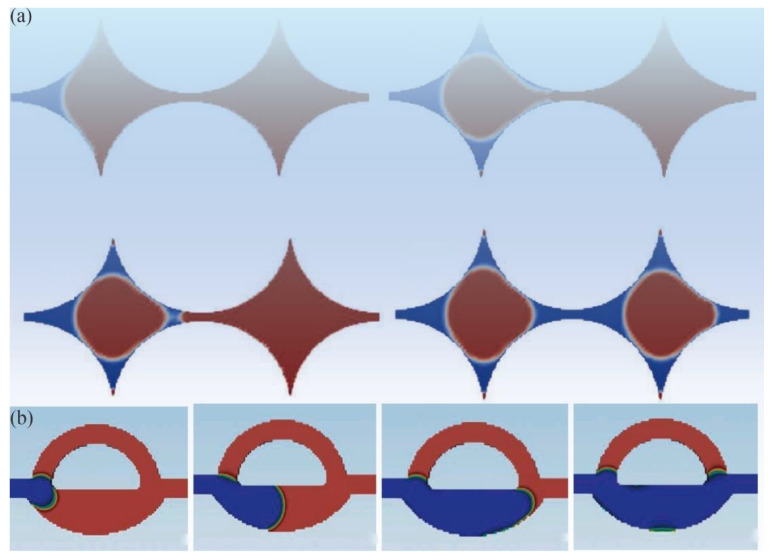
(**a**) Illustration of the wetting phase (red) invading pores saturated with the non-wetting phase (blue); (**b**) Schematic of oil phase entrapment (red) caused by water and gas bypassing at different pore volumes (PV) of injection (blue indicates the water or gas phase) [42]. (Copyright 2022, with permission from Elsevier).

**Figure 4 molecules-29-03978-f004:**
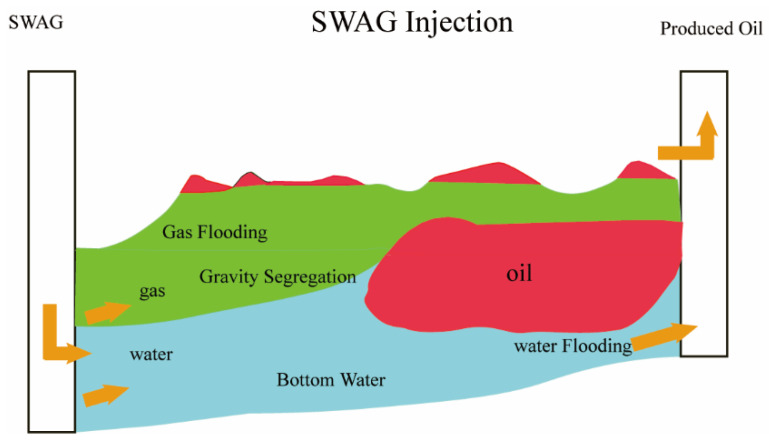
Schematic diagram of SWAG Injection.

**Figure 5 molecules-29-03978-f005:**
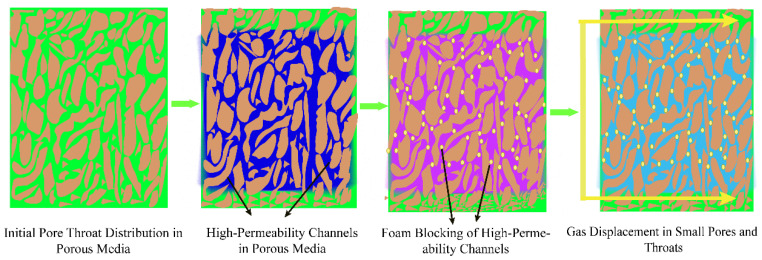
Schematic diagram of FAWAG injection technology.

**Figure 6 molecules-29-03978-f006:**
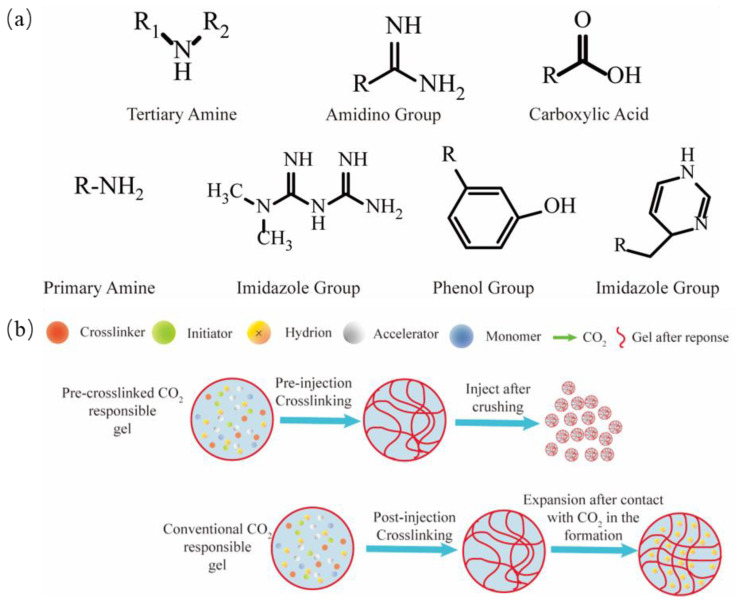
(**a**) CO_2_-responsive functional groups; (**b**) CO_2_-responsive gel mechanism (modified after Ding Y et al. [125]).

**Figure 7 molecules-29-03978-f007:**
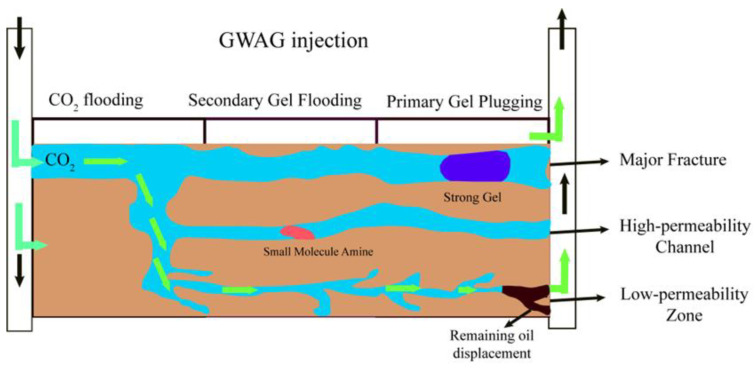
Schematic diagram of the multi-stage GWAG technology.

**Figure 8 molecules-29-03978-f008:**
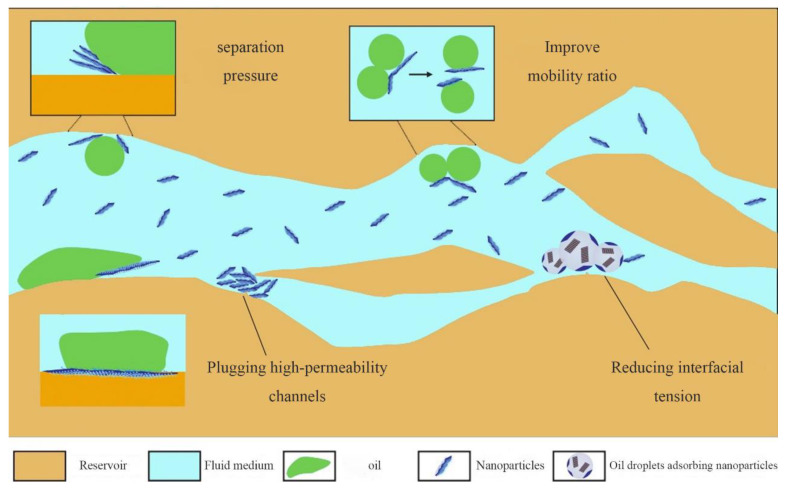
Schematic diagram of NWAG injection technology.

**Table 1 molecules-29-03978-t001:** WAG flooding: key factors in controlling the sequence and methods for expanding the sweep volume.

Key Factors	Difficulty of Control	Sequence and Importance of Control	Main Control Methods	Control Mechanisms
Mobility ratio	Easy	Critical and prioritized control	Adding chemicals or altering injection methods in water–gas	Lowering the mobility ratio and stabilizing displacement fronts
Reservoir properties	Moderately challenging	Significant and targeted control	Adding chemicals to water–gas or adjusting injection-production regimes	Physically or chemically blocking fractures or high-permeability channels
Field injection-production regime	Relatively easy	Significant and targeted control	Modifying water–gas slug ratio, injection pressure, injection rate, and injection method	Reducing water–gas mobility ratio, stabilizing displacement front, and improving heterogeneity
Hysteresis and relative permeability	Moderately difficult	Significant and generally applicable control	Introducing chemical agents or adjusting injection-production regimes	Altering wettability, reducing gas phase permeability
Gravity segregation	Moderately difficult	Significant and generally applicable control	Control of injection-production regimes	Blocking vertical high-permeability channels to reduce vertical gas migration

**Table 2 molecules-29-03978-t002:** The common two-phase hysteresis models.

Two-Phase Hysteresis Models	Applicability to Fluid Phases and Rock Systems	Rock Wettability	Reversibility of Hysteresis Curves	Limitations and Drawbacks	Key Formulas	Ref
Killough model	Gas–water systems	Water-wet system	reversible	It performs well in predicting hysteresis effects in gas–water systems but fails to accurately predict hysteresis effects in gas–oil systems. As petroleum saturation increases, it underestimates gas relative permeability due to the internal inclusion of the Land model, resulting in poor hysteresis predictions for oil–water mixed-wet systems.	$K_{rN}^{Im}(S_{N})=K_{rN}^{Dr}(S_{N}^{Hyst})\cdot [\frac{K_{rN}^{Exp}(S_{N}^{Norm})-K_{rN}^{Exp}(S_{Nr}^{Max})}{K_{rN}^{Exp}(S_{N}^{Max})-K_{rN}^{Exp}(S_{Nr}^{Max})}]$ $S_{Nr}=\frac{S_{N}^{Hyst}}{1+C*S_{N}^{Hyt}}$ $C=\frac{1}{S_{Nr}^{Max}}-\frac{1}{S_{N}^{Max}}$	[64]
Carlson model	Oil–water systems	Water-wet system	reversible	It performs well in predicting hysteresis in oil–water mixed-wet systems, but shows poorer predictions for hysteresis effects in gas–oil and gas–water systems during the imbibition process of the non-wetting phase. Additionally, the model’s prediction performance is affected by the absence of hysteresis effects in the wetting phase.	$\frac{1}{S_{mwr}}-\frac{1}{S_{mwi}}=C$ $S_{nwf}=\frac{1}{2}[(S_{nw}-S_{nwr})+\sqrt{\left(S_{nw}-S_{nwr}{)}^{2}+\frac{4}{C}(S_{nw}-S_{nwr}\right)}]$ $k_{mw}^{I}(S_{nw})=k_{rmw}^{D}(S_{nwf})$	[65]
Beattie model	Gas–oil systems	Water-wet system	irreversible	The model performs well in predicting hysteresis effects in gas–oil systems. However, the uncertainty and multiplicity of parameter (n) arise from fitting curves to estimate it through the first imbibition and drainage cycles. Additionally, a major drawback of the model is the absence of a trapping model.	$k_{rw}^{*}=k_{rwi}^{*}-[\frac{k_{rwi,P}^{*}-k_{rw,P}^{*}}{k_{rwi,p}^{*}-k_{rwd,P}^{*}}]{\left[\frac{1-S_{w}^{*}}{1-S_{w,P}^{*}}\right]}^{n}(k_{rwi}^{*}-k_{rwd}^{*})$ $k_{ro}^{*}=k_{rod}^{*}-[\frac{k_{rod,P}^{*}-k_{r0,P}^{*}}{k_{rod,P}^{*}-k_{roi,P}^{*}}]{\left[\frac{S_{w}^{*}}{S_{w,P}^{*}}\right]}^{n}(k_{rod}^{*}-k_{roi}^{*})$	[73]
Kjosavik model	Gas–oil systems	Mixed-wet system	irreversible	The model extends relative permeability relationships for water-wet and oil-wet systems to mixed-wet rocks. However, due to its inclusion of the Land trapping model, the model has inherent limitations for non-water-wet systems.	$k_{rwi}[j](S_{w})=k_{rwi}^{0}[j]k_{rwi}^{1}(S_{w})+k_{rwi}^{t}[j]$ $k_{roi}[j](S_{w})=k_{roi}^{0}[j]k_{roi}^{1}(S_{w})+k_{roi}^{t}[j]$	[68]
Spiteri model	Pore network model	Mixed-wet system	irreversible	This trapping model is applicable to multiple wetting systems and performs well in predicting hysteresis effects in oil–water systems.	$S_{ot}=\alpha S_{oi}-\beta S_{oi}^{2}$ $\Delta S_{o}=S_{ot}(S_{oi})-S_{ot}(S_{of})-\gamma [S_{o}-S_{ot}(S_{oi})](S_{o}-S_{oi})$ $S_{of}=\frac{1}{2\beta }[(\alpha -1)+\sqrt{\left(\alpha -1{)}^{2}+4\beta [S_{0}-S_{ot}+\gamma (S_{o}-S_{ot})(S_{o}-S_{oi})\right]}]$	[70]

**Table 3 molecules-29-03978-t003:** The common three-phase hysteresis models.

Three-Phase Hysteresis Models	Analysis	Rock Wettability	Limitations and Drawbacks	Key Formulas	Ref
Stone 1 model	The model does not account for hysteresis effects, and while it accurately predicts relative permeability at high oil saturations, it tends to overestimate the relative permeability of the oil phase as the oil saturation decreases.	Water-wet system	The model predictions are based on imbibition hysteresis curves under different initial conditions, rather than on drainage and imbibition hysteresis curves during cyclic periods. In mixed-wet or weakly water-wet systems, the prevailing models’ assumption that the relative permeabilities of the wetting phase (water) and the non-wetting phase (gas) are exclusively determined by their own saturations is not reflective of actual reservoir conditions. In these systems, krw and krg are dependent on both saturations. This fundamental discrepancy results in a significant divergence between the model predictions and the experimental outcomes for mixed-wet or weakly water-wet systems.	$k_{r0}(S_{w},S_{g})=\frac{s_{o}k_{row}(S_{w})k_{rog}(S_{g})}{k_{row}(1-S_{w1})(1-s_{g1})}$ $s_{w1}=\frac{S_{w}-S_{wc}}{1-S_{wc}-S_{gc}-S_{or}}$ $s_{g1}=\frac{S_{g}-S_{gc}}{1-S_{wc}-S_{gc}-S_{or}}$ $s_{o1}=\frac{S_{o}-S_{or}}{1-S_{wc}-S_{gc}-S_{or}}$	[77]
Stone 2 model	The model does not account for hysteresis effects, and it may underestimate the relative permeability of the oil phase when predicting three-phase permeability.	Water-wet system		$k_{ro}(S_{w},S_{g})=k_{row}^{0}[\frac{k_{row}\left(S_{w}\right)}{k_{row}^{0}}+k_{rw}(S_{w})][\frac{k_{rog}\left(S_{g}\right)}{k_{row}^{0}}$ $+k_{rg}(S_{g})]-[k_{rw}(S_{w})+k_{rg}(S_{g})]\}$	[78]
Baker model	The model assumes that water and gas are completely segregated within the porous medium, while the oil phase is uniformly distributed.	Water-wet system		$k_{ro}(S_{w},S_{g})=\frac{(S_{w}-S_{wc})k_{row}^{*}+(S_{g}-S_{gc})k_{rog}^{*}}{\left(S_{w}-S_{wc})+(S_{g}-S_{gc}\right)}$ $k_{row}^{*}=k_{row}(1-S_{o})=k_{row}(S_{w}+S_{g})$ $k_{rog}^{*}=k_{rog}(1-S_{wc}-S_{o})=k_{rog}(S_{g}+S_{w}-S_{wc})$	[79]
Linear model	The model assumes a linear relationship between the permeabilities of the oil–water and oil–gas curves, allowing for the determination of the three-phase oil relative permeability.	Water-wet system		$k_{ro}\left(S_{w},S_{g}\right)=k_{row}\left(S_{w}^{ow}\right)=k_{rog}\left(S_{g}^{og}\right)$ $\frac{S_{w}-S_{w}^{ow}}{S_{w}-S_{wc}}=\frac{S_{g}-S_{gc}}{S_{g}-S_{g}^{og}}$	[80]

**Table 4 molecules-29-03978-t004:** Major Research Advances in SWAG Technology.

Researchers, Year	Porous Medium Type	Research Method	Key Findings	Ref
Stephenson DJ, 1993	sandstone	Joffre Viking oil field	SWAG in a 1:1 ratio enhances sweep efficiency compared to conventional CO_2_-WAG and continuous CO_2_ injection.	[15]
Ma Yunfei, 2015	sandstone	laboratory core displacement experiment.	The coefficient of fluid mobility reduction in SWAG is inversely proportional to core permeability, which is a key indicator of SWAG’s capability to expand sweep volume.	[84]
Berge L I, 2002	sandstone	Siri oil field field experiments and numerical simulation (Eclipse)	The swept volume of SWAG is influenced by injection and production system parameters, specifically injection pressure and the gas-to-water ratio. Decreased injection capacity reduces the range of gas displacement, with injection capacity dependent on the gas-to-water ratio under fracturing pressures.	[87]
Stone, 2004	limestone	Laboratory model experiments and numerical simulation (UTCOMP)	SWAG is superior to conventional CO_2_-WAG and continuous CO_2_ injection modes because it better controls gas mobility, resulting in relatively higher oil recovery rates.	[88]
M.Jamshidnezhad, 2008	porous network model	numerical simulation (CMG).	Factors such as injection rates of water and gas, the placement of injection wells, and reservoir heterogeneity significantly influence gas mobility and sweep range, with minimal impact on oil recovery rates.	[82]

**Table 5 molecules-29-03978-t005:** Major Research Advances in PWAG and GWAG Technology.

Researchers, Year	Porous Medium Type	Synthetic Ingredient	Research Methods	Key Findings	Ref
W. Li, 2014	Pore Network Mode	Not mentioned	Numerical Simulation (CMG) of TR59 Block in North Burbank Oilfield	A case study of the TR59 block demonstrates that the incremental recovery rate using PWAG is 20%, which is 12% higher than that achieved with WAG. Heterogeneous formations with high permeability variation coefficients are the optimal choice for PWAG flooding.	[19]
Yang Y, 2018	Pore Network Mode	Not mentioned	Numerical Simulation (CMG) of Heterogeneous Heavy Oil Reservoirs in Liaohe Oilfiel	The recovery rate using PWAG technology is 45% higher than that of polymer flooding and 57% higher than that of WAG. It significantly reduces the water cut and gas–oil ratio.	[126]
ZHAO, 2015	sandstone	Modified Starch Gel and Ethylenediamine	Core Flooding Experiments and Field Trials	When the permeability ratio is less than or equal to 100, the in situ gel generated from the reaction of ethylenediamine with CO_2_ effectively seals high-permeability layers. This process enhances CO_2_ flooding efficiency and boosts the total recovery rate by over 20%.	[127]
Hao H, 2016	sandstone	Modified Starch Gel, Acrylamide Monomer Crosslinker, and Ethylenediamine	Three-dimensional radial flow model and core displacement experiments	Injection of high-strength gel and ethylenediamine combined with CO_2_-WAG flooding effectively mitigates gas channeling in large fractures and high-permeability channels. This integrated approach results in a substantial 15.09% increase in overall recovery rates	[128]
Luo et al., 2022	sandstone	N-erucamidopropyl- N, N-Dimethylamine	numerical simulation (CMG).	The recovery rate of UC22AMPM solution WAG increased by 8% compared to WAG.	[129]
Luo X et al., 2021	sandstone	F127-g-PDMAEMA	Core flooding test	Compared to conventional WAG, HAPM solution WAG increases total recovery by 21% and expands the gas displacement volume.	[130]
Mingwei Zhao, 2023	sandstone	Sodium salicylate, acetone, sodium fluoride, erucic acid N, N-dimethyl-1,3- propane diamine (99%).	Dual-tube core parallel flooding test and CO_2_ plugging experiment	Low permeability, high surfactant concentration, and high injection volume are positively correlated with improved plugging performance. Gel systems enhance reservoir conformance, resulting in an 18.7% increase in recovery rate for low-permeability reservoirs.	[131]

**Table 6 molecules-29-03978-t006:** Comparison of Techniques for Enhancing Sweep Efficiency in CO_2_-WAG Flooding with Chemical and Physical Assistance.

Method	Influencing Factor	Advantage	Disadvantage	Field Application
WAG	Reservoir factors (temperature, pressure, thickness, porosity, permeability, saturation, and heterogeneity); CO_2_ rheological properties and density; composition and viscosity of crude oil; hysteresis, Gravity Segregation Injection parameters (injection rate, injection pressure, and slug ratio); injection scheme; and spacing between injection and production wells.	It enhances CO_2_ utilization and increases crude oil recovery, controls the mobility ratio, reduces gas channeling, expands sweep efficiency, reduces the mobility ratio, and maintains CO_2_ mixing with crude oil.	Controlling gas channeling in strong heterogeneous and fractured oil reservoirs may prove ineffective; corroded pipelines are unable to alleviate the phenomenon of gravitational differentiation, which causes the water lock effect.	Yes
SWAG	Reservoir factors (temperature, pressure, thickness, porosity, permeability, saturation, and heterogeneity); CO_2_ rheological properties and density; composition and viscosity of crude oil; hysteresis effect and gravity segregation; injection parameters (injection rate and injection pressure); injection scheme; and spacing between injection and production wells.	It enhances CO_2_ utilization and increases crude oil recovery; controlling gas channeling and fingering has shown significant effects.	Corrosion of the casing and wellbore, instability of the water–gas front, inability to alleviate the phenomenon of gravitational differentiation, and causing the water lock effect.	Yes
FWAG	CO_2_ rheological properties and density; composition and viscosity of crude oil; injection scheme; type and concentration of surfactants; molecular structure; injection parameters (injection rate and injection pressure); reservoir factors (temperature, pressure, thickness, porosity, permeability, saturation, and heterogeneity); and chemical agent formulation and system.	Relieves sticky fingering; relieves gravity differentiation; relieves early breakthroughs; reduces interfacial tension; changes wettability; easy to inject; and prevents and controls sedimentation.	The cost is high, the foam stability is poor and prone to defoaming, it is sensitive to temperature and pressure, and its effective range is short.	Yes
PWAG	Reservoir factors (temperature, pressure, thickness, porosity, permeability, saturation, and heterogeneity); CO_2_ rheological properties and density; composition and viscosity of crude oil; injection scheme; injection parameters (injection pressure, injection rate, and injection fluid concentration), and chemical agent formulation and system.	It can effectively block large fractures and high-permeability channels in highly heterogeneous oil reservoirs.	The high cost, difficulty in injecting, limited effective range, and potential damage to the reservoir pose significant concerns.	Yes
GWAG	Reservoir factors (temperature, pressure, thickness, porosity, permeability, saturation, and heterogeneity); CO_2_ rheological properties and density; composition and viscosity of crude oil; injection scheme; injection parameters (injection pressure, injection rate, and injection fluid concentration); and chemical agent formulation and system.	Good injectivity can improve reservoir heterogeneity, control mobility ratio, and mitigate gas channeling.	Sensitive to reservoir conditions, and strict chemical agent formulation and system	No
NWAG	Reservoir factors (temperature, pressure, thickness, porosity, permeability, saturation, and heterogeneity); CO_2_ rheological properties and density; composition and viscosity of crude oil; injection scheme; nanoparticle type; particle size; hydrophilicity; and concentration.	Reduces the mobility ratio; prevents and controls asphaltene precipitation; changes the wettability of rocks; reduces interfacial tension; and improves CO_2_ rheological properties.	Nanoparticles are prone to coalescence, blocking the roar channel, and failing; large particle sizes can damage the reservoir.	No

## Data Availability

All data generated or analyzed during this study are contained within the article.

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
