# Peer review of "Chemical-Assisted CO2 Water-Alternating-Gas Injection for Enhanced Sweep Efficiency in CO2-EOR"

_molecules, 2024, doi:10.3390/molecules29163978_

Round 1

Reviewer 1 Report

Comments and Suggestions for Authors

The authors reviewed the application and research progress of chemical-assisted CO2 water-alternating-gas (CO2-WAG) technology in CO2-EOR. They analyzed key factors affecting CO2-WAG flooding efficiency, such as mobility ratio, reservoir properties, hysteresis effects, relative permeability and injection-production scheme. The mathematical model of WAG flooding. And they explored methods to optimize WAG flooding efficiency, including simultaneous water and gas injection, foam-assisted WAG, polymer-gel-assisted WAG, and nanoparticle-assisted WAG. To enhance the comprehensiveness of the review, I suggest that the authors consider supplementing the following point.

In the polymer-gel-assisted WAG section, the authors mentioned that it can reduce CO2 gas channeling and fingering by blocking high-permeability layers and fractures. Additionally, some polymers also act as thickeners, enhancing the viscosity of CO2 to reduce fingering and channeling. Recently, there are studies on the thickening effects of hydrocarbon polymers, fluoropolymers, and silicon-containing polymer thickeners on CO2. It is hoped that the authors can supplement some content on the effects of different thickeners on CO2-WAG.

Comments on the Quality of English Language

I have not any comments about language.

Reviewer 2 Report

Comments and Suggestions for Authors

The article systematically investigates the impact of various factors on CO2-WAG flooding sweep efficiency and proposes regulatory strategies., reviews the existing main CO2-WAG mathematical models, and summarizes and discusses the effects of different hysteresis models on WAG flooding production. Addressing the current issues of severe gas channeling and CO2-WAG flooding regulation failure in heterogeneous and fractured reservoirs, the study analyzes and summarizes the latest advancements in chemically assisted CO2-WAG flooding for enhancing sweep efficiency. This includes key mechanisms, control factors, and future research directions, representing significant progress in this field. But Minor Revision revisions are required before publication. Followings are my comments.

1Building on a thorough review of the main issues associated with existing CO2-WAG techniques and the previous work in this area, the authors should highlight the advantages of chemically assisted CO2-WAG flooding for enhancing sweep efficiency in the introduction section. This will align with the article's title and underscore the significance of this technology.

2Although Tables 2 and 3 provide the names of the models, they do not include specific references for these models. Additionally, the meanings of the variables in the formulas within the tables are not explained. The authors should provide the appropriate references and explanations for the relevant variables.

3The title of Section 4.3 is ambiguous and should logically be revised to "Polymer or Gel-Assisted Expansion of Water-Alternating-Gas (WAG) Techniques." Additionally, the authors have not explained the differences between the three different techniques and their respective application scenarios. This lack of clarity may confuse readers. The authors should reconsider the section title and clearly articulate the distinctions and contexts for each technique.

4The table headings and formatting in the article are inconsistent. The authors should reformat the content to ensure that the tables are presented in a uniform and standardized manner.

5The title of the third section is "Establishment of WAG flooding mathematical models," but most of the content in this section discusses hysteresis models. The authors should ensure that the title aligns with the content to help readers easily understand the correspondence between the title and the section's content.

Reviewer 3 Report

Comments and Suggestions for Authors

In their paper, the authors present an updated and comprehensive review of methods for improving the efficiency of CO2 oil recovery methods using water-alternating-gas injection (WAG) technology. The paper incorporates the latest scientific findings and applies a wide range of research methods, from experiments to pore-scale modeling. The authors discuss the challenges faced when implementing WAG injection in heterogeneous and fractured formations and propose strategies and methods to overcome them. The authors also draw attention to the operational challenges of WAG projects and propose measures to address them. The paper is positive and does not raise any fundamental comments.

IMPORTANT ERROR !

1. In lines 780-783, the reference affects 3 reference points. Lines 10-11 affect two reference points, 15-16 affect 2 reference points, etc. All references must be checked!!!!!

2. Some links have incomplete data, for example, lines 813, 814, 815, 795, etc.

Notes:

1. Line 80. A comma is missing

2. Lines 91-91. The sentence is too long, construct the phrase differently.

3. Line 149. Text 7%-8% must be supplemented with a link.

4. Check that all variables in the formulas are disclosed in the text.

5. Line 185. It is necessary to supplement with a link to the work doi.org/10.1109/SCM50615.2020.9198816

6. Lines 269, 274 formulas must be separated by "; "

7. Tables 2 and 3. variables are not disclosed. Either indicate where the formulas are taken from or add a description of the formulas.

8. Figure 4. Text "oil-water interface". Are you sure the word "interface" should be there?

9. Line 708. Possible spelling error.

Round 2

Reviewer 1 Report

Comments and Suggestions for Authors

The authors have substantially revised the original submission entitled "Chemical-assisted CO2 water-alternating-gas injection for Enhanced sweep efficiency in CO2-EOR". They have provided detailed responses toward the suggestions from the reviewers and in my point of view, the answers are justified by the modification of the manuscript. I recommend this work can be considered for publication in Molecules journal.